# Atovaquone and Pibrentasvir Inhibit the SARS-CoV-2 Endoribonuclease and Restrict Infection In Vitro but Not In Vivo

**DOI:** 10.3390/v15091841

**Published:** 2023-08-30

**Authors:** Troy von Beck, Luis Mena Hernandez, Hongyi Zhou, Katharine Floyd, Mehul S. Suthar, Jeffrey Skolnick, Joshy Jacob

**Affiliations:** 1Emory Vaccine Center, Emory National Primate Research Center, Emory University, 954 Gatewood Road, Atlanta, GA 30329, USA; tvonbec@emory.edu (T.v.B.); luis.mena@pitt.edu (L.M.H.); katharine.floyd@emory.edu (K.F.); mehul.s.suthar@emory.edu (M.S.S.); 2Center for the Study of Systems Biology, School of Biological Sciences, Georgia Institute of Technology, 950 Atlantic Drive, Atlanta, GA 30332, USA; hongyi.zhou@biology.gatech.edu (H.Z.); jeffrey.skolnick@biology.gatech.edu (J.S.); 3Department of Pediatrics, Division of Infectious Diseases, Emory University School of Medicine, Atlanta, GA 30322, USA

**Keywords:** SARS-CoV-2, nsp15, endoribonuclease, antiviral

## Abstract

The emergence of SARS-CoV-1 in 2003 followed by MERS-CoV and now SARS-CoV-2 has proven the latent threat these viruses pose to humanity. While the SARS-CoV-2 pandemic has shifted to a stage of endemicity, the threat of new coronaviruses emerging from animal reservoirs remains. To address this issue, the global community must develop small molecule drugs targeting highly conserved structures in the coronavirus proteome. Here, we characterized existing drugs for their ability to inhibit the endoribonuclease activity of the SARS-CoV-2 non-structural protein 15 (nsp15) via in silico, in vitro, and in vivo techniques. We have identified nsp15 inhibition by the drugs pibrentasvir and atovaquone which effectively inhibit SARS-CoV-2 and HCoV-OC43 at low micromolar concentrations in cell cultures. Furthermore, atovaquone, but not pibrentasvir, is observed to modulate HCoV-OC43 dsRNA and infection in a manner consistent with nsp15 inhibition. Although neither pibrentasvir nor atovaquone translate to clinical efficacy in a murine prophylaxis model of SARS-CoV-2 infection, atovaquone may serve as a basis for the design of future nsp15 inhibitors.

## 1. Introduction

Severe acute respiratory syndrome coronavirus 2 (SARS-CoV-2), the causative agent of the coronavirus disease of 2019 (COVID-19), continues to spread globally and cause significant morbidity and mortality among unvaccinated individuals and those with weakened immune systems [1]. Furthermore, control of the virus has been complicated by the expansion of novel variants possessing mutations that enhance transmissibility and erode the protection provided by the original A.1 lineage-based vaccines, convalescent plasma, and monoclonal therapies [2,3]. Contrary to the observed evolution of SARS-CoV-2 antibody escape variants, susceptibility to the small molecule antiviral remdesivir has remained consistent overtime, despite the sporadic appearance of remdesivir resistant viruses in hospitalized patients [4,5]. This trend likely reflects the lack of selection for remdesivir-resistant strains outside the hospital environment and highlights the long-term value of identifying new small molecules targeting conserved viral proteins.

To complement the antagonism of viral RNA synthesis and polypeptide processing by remdesivir, molnupiravir, and ritonavir-boosted nirmatrelvir (Paxlovid), we focused on identifying inhibitors of the viral endoribonuclease encoded by the non-structural protein 15 (nsp15). This protein is conserved among all mammalian coronaviruses and functions to regulate the accumulation and localization of viral double-stranded RNA (dsRNA). Nsp15 cleaves RNA at unpaired pyrimidine bases, with a strong preference for uridine over cytidine [6,7,8]. This nuclease activity functions to limit cytosolic dsRNA and thereby antagonizes intracellular pattern recognition receptors (PRRs) specific for dsRNA and 5′ poly(U) sequences, including MDA5, PKR, and OAS/RNase L [9,10,11]. Deficiency for nsp15 has been linked to a reduced viral fitness across multiple coronaviruses including SARS-CoV-1, human coronavirus 229E (HCoV-229E), infectious bronchitis virus (IBV), mouse hepatitis virus (MHV), and porcine epidemic diarrhea virus (PEDV) [9,10,11,12,13,14]. In those studies, the loss of viral fitness correlated with the early induction of a type I interferon response and the rapid apoptosis of infected cells. Furthermore, in the context of MHV, knockout mutations in nsp15 were not found to increase total cellular dsRNA content but did result in a greater proportion of “free” dsRNA foci not sequestered in the replication complex associated double-membraned vesicles (DMVs) [10]. The conserved dependence of SARS-CoV-1 and other distantly related coronaviruses on nsp15 for immune evasion strongly supports the existence of a similar dependence for SARS-CoV-2.

Due to the pressing need for SARS-CoV-2 antiviral therapeutics, there is a heightened interest in repurposing drugs approved by the United States Food and Drug Administration (USFDA) for the treatment of COVID-19 patients. This approach can accelerate the drug discovery pipeline by limiting the selection of potential therapeutics to those with known safety and tolerability profiles [15]. Additionally, in silico analyses using algorithms like AutoDock, Dockthor, GOLD, or FRAGSITE can stratify the candidates for drug repurposing based on the strength of modeled interactions with a target protein [16,17,18].

In the following experiments, we applied the newly updated FRAGSITE2 virtual ligand screening algorithm to rank potential SARS-CoV-2 nsp15 inhibitors from the DrugBank compound library [19]. Top ranked drugs were screened for the inhibition of recombinant nsp15 enzymatic activity and the inhibition of SARS-CoV-2 and HCoV-OC43 infection in cell cultures. This yielded two candidate drugs, pibrentasvir (predicted by FRAGSITE2) and atovaquone (predicted by other in silico methods), which were MAVS-dependent and acted early during viral infections. However, in a murine infection model with a live SARS-CoV-2 virus, we found that neither atovaquone nor pibrentasvir were able to reduce viral replications or a symptomatic weight loss; while the origin of this difference in response is unclear, it might reflect problems with bioavailability.

## 2. Materials and Methods

### 2.1. In Silico Screening of FDA Approved Drugs for Nsp15 Binding Activity

Given the amino acid sequence of SARS-CoV-2 nsp15, FRAGSITE2 predicts its three-dimension (3D) structure using our previously developed method TASSER [20]. This method produced a structure with no major deviations from the experimentally determined protein data bank (PDB) 6VWW crystal structure of nsp15 (TM-score 0.90) and performed well in our methodology. The 3D structure of a single nsp15 monomer was then compared to a library of experimentally determined protein–ligand binding pockets from the PDB [21]. The top list of matched pockets is used to derive a profile representing the target pocket, which is then combined with the ligand profiles of the screened compound library to form a feature vector. We then utilized a boosted tree regression machine learning method to train a model on the DUD-E benchmarking set [22]. This model was then used to make new predictions by screening on the DrugBank database [23].

### 2.2. Growth and Purification of Recombinant SARS-CoV-2 Nsp15

To produce recombinant nsp15 for in vitro screening assays, a vector encoding the SARS-CoV-2 nsp15 with a c-terminal twin strep tag (Addgene, Watertown, MA, USA, Catalog#:141381) was transfected into HEK293T cells using the transporter 5 PEI transfection reagent (Polysciences, Warrington, PA, USA, Catalog#: 26008-5). The 72 h post-transfection adherent cells were collected by mechanical scraping, washed once in 1× PBS, and lysed in ice-cold 1× PBS containing 1% NP-40 and a 1× concentration of a Halt protease inhibitor cocktail (Thermo Fisher Scientific, Waltham, MA, USA, Catalog#: 78429). Recombinant nsp15 was column purified by strep-tag affinity (Zymo Research, Irvine, CA, USA, Catalog#: P2004). Purified nsp15 in the Strep-Elution Buffer (Zymo Research, Irvine, CA, USA, Catalog#: P2004-3-30) was diluted 1:1 with 100% glycerol and stored at −20 °C until use. Proper assembly of the recombinant nsp15 into homohexamers was confirmed by Western blot analysis following native polyacrylamide gel electrophoresis (Appendix A). For the Western blot analysis, recombinant nsp15 was detected by the polyclonal rabbit anti-Strep II tag antibody (Abnova, Taipei, Taiwan, Catalog#: PAB16601) and the anti-rabbit Ig HRP conjugated secondary antibody (SouthernBiotech, Birmingham, AL, USA, Catalog#: 4030-05). In subsequent experiments, SARS-CoV-2 nsp15 bearing an N-terminal 6x-his tag was purchased from a commercial supplier (Novus Biologicals, Centennial, CO, USA, Catalog#: NBP3-07082) and stored frozen in PBS without the addition of 50% glycerol.

### 2.3. Library Screening for SARS-CoV-2 Nsp15 Inhibitory Activity

The direct inhibition of SARS-CoV-2 nsp15 nuclease activity by each inhibitor was measured in vitro using recombinant strep-tagged or his-tagged nsp15 and a 5′-FAM-dA-rU-dA-dA-TAMRA-3′ FRET probe as previously described [24]. Briefly, purified SARS-CoV-2 nsp15 was added to a concentration of 30 nM in the NendoU buffer (100 mM of NaCl, 20 mM of HEPES, pH 7.8) supplemented with 5 mM of MnCl_2_, 0.5µM of the FRET probe, and the inhibitor at the specified concentration on ice. For experiments with the strep-tagged nsp15, FAM fluorescence was recorded at 1 min intervals for 1 h on a Bio-rad CFX 96 real-time qPCR thermocycler (Bio-Rad, Hercules, CA, USA) at 30 °C using the SYBR channel. For experiments with the his-tagged nsp15, FAM fluorescence was recorded at 1 min intervals for 1 h on a LightCycler480 (Roche, Basel, Switzerland) at 37 °C using the SYBR channel. For analysis, the recorded fluorescence of each well after 10 min was normalized to the starting fluorescence and to a no-enzyme negative control well. The inhibition of nuclease activity was then calculated as the percent reduction in FAM fluorescence relative to the no-inhibitor positive control wells.

### 2.4. Evaluation of Drug-Induced Nsp15 Aggregation

The N-terminally his-tagged nsp15 (Novus Biologicals, Centennial, CO, USA, Catalog#: NBP3-07082) was diluted to 700 nM in the NendoU buffer containing 5 mM of MnCl_2_ with or without 100 µM of a drug additive. The control and drug-treated samples were then heated at 37 °C for 10 min to simulate the nuclease assay reaction conditions. As a positive control, an additional preparation of the control nsp15 without the drug was heated to 95 °C for 10 min to denature the protein and induce aggregation. Following incubation, the NativePAGE sample buffer was added to a 1× concentration, and the protein preparations were then separated on a 4–12% non-denaturing Bis-Tris polyacrylamide gel (Thermo Fisher Scientific, Waltham, MA, USA, Catalog#: NP0323BOX) using a light cathode buffer (Thermo Fisher Scientific, Waltham, MA, USA, Catalog#: BN2002). Proteins were then transferred to a PVDF membrane and washed twice with methanol followed by blocking with TBST containing 5% *w*/*v* non-fat dry milk for 1 h. Membranes were then stained with the primary anti-6x-his tag antibody (Thermo Fisher Scientific, Waltham, MA, USA, Catalog#: PA1-983B) diluted 1:1000 in TBST overnight. Membranes were then washed 3 times with TBST and stained with the anti-rabbit Ig HRP conjugated secondary antibody (SouthernBiotech, Birmingham, AL, USA, Catalog#: 4030-05) diluted 1:10,000 in TBST for 2 h. After 3 additional washes, protein bands were detected by chemiluminescence using the SuperSignal West Femto Maximum Sensitivity Substrate (Thermo Fisher Scientific, Waltham, MA, USA, Catalog#: 34094).

### 2.5. Cell Lines and Viruses

HEK293T cells were purchased from ATCC (ATCC, Manassas, VA, USA, Catalog#: CRL-3216). A549 cells expressing human ACE2 and TMPRSS2 were purchased from InvivoGen (InvivoGen, San Diego, CA, USA, Catalog#: a549-hace2tpsa). Wild-type A549 and A549 MAVS^−/−^ were provided by the lab of Dr. Horner (Duke University). VeroE6-TMPRSS2 cells were kindly provided by Barney Graham (Vaccine Research Center, NIH, Bethesda, MD, USA). All cell lines were cultured in DMEM (Thermo Fisher Scientific, Waltham, MA, USA, Catalog#: 12-614Q) with 10% heat-inactivated fetal bovine serum (Gibco, Billings, MO, USA, Catalog#: A5256701), 2 mM L-glutamine (Quality Biological, Gaithersburg, MD, USA, Catalog#: 118-084-721), and 1× concentrations of penicillin, streptomycin, and amphotericin (PSA) (Quality Biological, Gaithersburg, MD, USA, Catalog#:120-096-711).

The B.1.351 variant (GISAID: EPI_ISL_890360) was provided by Andy Pekosz of John Hopkins University, Baltimore, MD. Both the SARS-CoV-2 B.1.351 and A.1 (nCoV/USA_WA1/2020) viral stocks were grown on VeroE6-TMPRSS2 cells, and viral titers were determined by plaque assays on the VeroE6-TMPRSS2 cells. The VeroE6-TMPRSS2 cells were cultured in complete DMEM with puromycin at 10 mg/mL (Gibco, Catalog#: A11138-03). HCoV-OC43 (VR-1558) was obtained from ATCC (Manassas, VA, USA) and grown on A549 WT cells. The HCoV-OC43 stocks were titered by focus forming assay.

### 2.6. Coronavirus Infection Inhibition Assay

For the infection assays, 96-well plates containing 2 × 10^4^ A549 hACE2 TMPRSS2 or A549 WT cells were used for the SARS-CoV-2 and HCoV-OC43 viruses, respectively. Cells were plated and grown overnight in complete DMEM at 37 °C and 5% CO_2_ prior to infection with an MOI of 0.1 of SARS-CoV-2 or HCoV-OC43 in 50 µL of unsupplemented DMEM. Cells were infected at either 37 °C or 33 °C for SARS-CoV-2 and HCoV-OC43, respectively. A total of 2 h after infection, 2× the concentrations of each inhibitor diluted in DMEM with 2% FBS, 2× PSA, and 2× L-glutamine were added. Infected cells were then incubated for 48 h at 5% CO_2_ and 37 °C or 24 h at 5% CO_2_ and 33 °C for SARS-CoV-2 and HCoV-OC43, respectively. The reduced growth time and temperature for HCoV-OC43 was used to compensate for its faster growth time in A549 cells and its preference for upper respiratory tract conditions.

Infected cells were quantified by focus forming assay. Briefly, infected cells were fixed with 2% paraformaldehyde and permeabilized by 1× PBS containing 0.1% saponin and 0.1% FBS. For SARS-CoV-2, the viral spike protein was detected by a human isotype CR3022 antibody (Abcam, Boston, MA, USA Catalog#: ab273073) and a goat anti-human Ig HRP conjugated secondary (SouthernBiotech, Catalog#: 2045-05). For HCoV-OC43, the viral nucleocapsid was detected by an anti-HCoV-OC43 nucleocapsid mouse monoclonal antibody (Millipore-Sigma, St. Louis, MO, USA, Catalog#: MAB9013) and a goat anti-mouse Ig HRP conjugated secondary (SouthernBiotech, Birmingham, AL, USA, Catalog#: 1030-05). Plates were developed for 10 min by incubation with TruBlue Peroxidase Substrate (SeraCare, Milford, MA, USA, Catalog #50-78-02). Images were collected on an Immunospot CTL instrument (Cleveland, OH, USA) and analyzed by an in-house script to quantify the area of infection. Plotted results represent the percentage of area of the cell monolayer infected relative to the control wells.

### 2.7. Cell Viability Assay

2 × 10^4^ A549 WT cells were plated and grown overnight in complete DMEM at 37 °C and 5% CO_2_ prior to the replacement of media with inhibitors diluted in 100 µL of 1% FBS containing DMEM with 1× PSA and 1× L-glutamine. Cells were incubated at 33 °C for 24 h before the addition of 20 µL of CellTiter 96-Aqueous One (Promega, Madison, WI, USA, Catalog#: PAG3580). Cells were then incubated at 33 °C for 1 h, followed by measurements of the absorbance at 450 nm.

### 2.8. Flow Cytometry of HCoV-OC43 Infected Cells

Confluent monolayers of A549 WT cells seeded in 12-well plates were infected with HCoV-OC43 at an MOI of 0.1 prior to the addition of inhibitors, FBS, L-glut, and PSA as in the focus forming assay. At 12 h post-infection, cells were washed once with 1× PBS (retained in the case of the non-adherent cells) and adherent cells were released by trypsinization. The adherent and non-adherent cell fractions were combined, pelleted, and resuspended in BD Cytofix/Cytoperm (BD Biosciences, Franklin Lakes, NJ, USA, Catalog#:554722). Cells were fixed and permeabilized for 20 min on ice followed by two washes with a BD perm/wash buffer (BD Biosciences, Franklin Lakes, NJ, USA, Catalog#: 554723). Cells were then stained with a mouse anti-HCoV OC43 N protein (Millipore-Sigma, St. Louis, MO, USA, Catalog#: MAB9013) diluted 1:200 in the perm/wash buffer for 30 min at room temperature. Cells were washed twice with the perm/wash buffer and then stained with an FITC conjugated goat anti-mouse Fab (SouthernBiotech, Birmingham, AL, USA, Catalog#: 1015-02) diluted 1:1000 in the perm/wash buffer. Cells were washed twice in 1× PBS containing 1% FBS and analyzed on a BD FACS Aria II flow cytometer (Franklin Lakes, NJ, USA) using the FITC channel.

### 2.9. Immunohistochemistry of Viral dsRNA in HCoV-OC43 Infected Cells

The A549 MAVS^−/−^ cells were seeded one day prior to infection on glass coverslips in complete DMEM media. Cells were then infected with HCoV-OC43 at an MOI of 0.1 for 2 h in an unsupplemented DMEM prior to the addition of inhibitors, FBS, L-glut, and PSA as in the focus forming assay. A total of 24 h later, cells were fixed in 4% paraformaldehyde in 1× PBS for 10 min and then permeabilized in an immunofluorescence perm/wash buffer containing 1× PBS, 0.5% saponin, 0.1% BSA, 5% normal goat serum, and 5% normal donkey serum. Cells were stained with a mouse anti-OC43 N protein (Millipore-Sigma, St. Louis, MO, USA, Catalog#: MAB9013) and a rabbit anti-dsRNA clone J2 (AbsoluteAntibody, Oxford, UK, Catalog#: Ab01299-23.0), each diluted 1:50 in the IF perm/wash buffer overnight at 4 °C. The unbound antibody was removed by washing in the perm/wash buffer. Cells were then stained with a secondary antibody PE conjugated donkey anti-rabbit Ig (BioLegend, San Diego, CA, USA, Catalog# 406421) and an AF488 conjugated goat anti-mouse Ig (LifeTechnologies, Carlsbad, CA, USA, Catalog#: A11001), diluted 1:300 in the perm/wash buffer for 1 h at 37 °C. Again, the unbound antibody was removed by washing in the perm/wash buffer, and excess saponin was then washed out with 1× PBS. Cell nuclei were then stained with Hoescht 33342 (Thermo Fisher Scientific, Waltham, MA, USA, Catalog#: 62249) and diluted 1:80 in 1× PBS for 10 min at 37 °C. Excess Hoescht 33342 was removed by washing in 1× PBS before mounting the coverslips to a glass slide with 100% glycerol. Slides were kept away from light and imaged on a fluorescence microscope within 3 h of mounting.

### 2.10. Infection of Mice with SARS-CoV-2 B.1.351

The C57BL/6J mice were purchased from Jackson Laboratories, Bar Harbor, ME, USA. All mice used in these experiments were females 8 weeks of age. The stock B.1.351 virus was diluted in 0.9% Normal Saline, USP (MedLine, Northfield, IL, USA, Catalog#: RDI30296). Mice were dosed with either 40 mg/kg of atovaquone or 24 mg/kg of pibrentasvir via daily intraperitoneal (i.p.) injections for 4 days, beginning one day prior to infection. Mice were anesthetized with isoflurane and infected intranasally with the virus (50 μL; 1 × 10^6^ PFU/mouse) in an animal biosafety level 3 (ABSL-3) facility. Mice were monitored daily for weight loss. All experiments adhered to the guidelines approved by the Emory University Institutional Animal Care and Use Committee.

### 2.11. Quantitative Reverse Transcription-PCR of Lung Tissues

At three days post-infection, mice were euthanized with an isoflurane overdose. One lobe of lung tissue was collected in an Omni Bead Ruptor tube filled with a Tripure Isolation Reagent (Roche, Basel, Switzerland, Catalog#: 11667165001). The tissue was homogenized using an Omni Bead Ruptor 24 instrument (5.15 ms, 15 s) and then centrifuged to remove debris. RNA was extracted using a Direct-zol RNA Miniprep Kit (Zymo Research, Irvine, CA, USA, Catalog#: R2051) and then converted to cDNA using a high-capacity reverse transcriptase cDNA kit (Thermo Fisher Scientific, Waltham, MA, USA, Catalog#: 4368813). To quantify RNA, the IDT Prime Time gene expression master mix was used with SARS-CoV-2 RDRP- and subgenomic-specific TaqMan gene expression primer/probe sets as previously described [25,26]. All qPCRs were performed in 384-well plates and run on a QuantStudio5 qPCR system.

## 3. Results

### 3.1. Predicted Nsp15 Binding Drugs Inhibit Nuclease Activity In Vitro

To identify drugs with the potential for nsp15 binding and inhibitory activity, we first performed an in silico screen of compounds contained in the DrugBank database using our recently developed FRAGSITE2 methodology. After a manual review of the predicted compounds to remove those not approved for human use by the USFDA, we compiled a small list of highly ranked drugs (Table 1). These drugs were predicted to interact with one of two binding pockets on the N-terminal oligomerization domain of nsp15. Oritavancin was the top hit predicted for binding pocket 1 which comprises the main interface between opposing monomers of nsp15 in the hexameric assembly (Figure 1A, Appendix A). Rifamixin was the top hit for binding pocket 2, which is composed of residues from both the N-terminal oligomerization domain and the middle-domain of nsp15 (Figure 1B, Appendix A). Rifamixin, rifapentine, and everolimus are predicted to bind the same pocket, although rifamixin also has a predicted interaction with a valine residue in position 31. Beyond our own predictions, we also selected several compounds predicted by previous nsp15 docking studies to bind the catalytic site, which included atovaquone, paritaprevir, glisoxepide, and idarubicin (Table 2) [27,28].

To validate our in silico predictions, we produced recombinant nsp15 bearing a C-terminal twin strep II epitope tag (Appendix A) and confirmed its cleavage of a small fluorescent RNA probe in vitro and its sensitivity to inhibition by the known RNase inhibitor benzopurpurin (Figure 2A) [29]. The in silico predicted drugs were then screened using the same nuclease assay for their ability to inhibit cleavage at a concentration of 100 µM (Figure 2B). As expected, benzopurpurin effectively prevented substrate RNA cleavage, as did idarubicin, pibrentasvir, and atovaquone. Conversely, cyanocobalamin, desmopressin, rifamixin, and rifapentine strongly suppressed the fluorescent signal generated by the substrate cleavage and could not be evaluated by this assay. Similarly, oritavancin was found to non-specifically induce the formation of insoluble aggregates of nsp15 and therefore was not a true inhibitor (Appendix A). In subsequent experiments, we also evaluated the ability of the top performing drugs benzopurpurin, pibrentasvir, atovaquone, and idarubicin to inhibit a commercially sourced SARS-CoV-2 nsp15 bearing an N-terminal his-tag tested at a physiological 37 °C (Figure 2B). Overall, the results were in agreement with the previous assay, although the reported inhibitory activity was reduced for each drug. Using the his-tagged nsp15, a dose–response experiment was also performed to evaluate the activity of each drug when serially diluted down to a concentration of 5 µM (Figure 2C).

### 3.2. Nsp15 Inhibitors Restrict SARS-CoV-2 Infection In Vitro

The candidate drugs benzopurpurin, pibrentasvir, atovaquone, and idarubicin were further investigated for their ability to restrict SARS-CoV-2 infection of A549 alveolar epithelial cells overexpressing the human ACE2 receptor and TMPRSS2 protease (A549 hAT) (Figure 3A). Consistent with previous investigations of SARS-CoV-1, benzopurpurin was incapable of restricting the SARS-CoV-2 infection of A549 hAT2 cells [29]. By comparison, atovaquone effectively inhibited the infection with an EC50 of 2.46 µM, and pibrentasvir inhibited greater than 90% of the infection even at the lowest concentration of 0.625 µM. Neither drug was observed to alter the cellular morphology or metabolic activity (Appendix A). Idarubicin is a genotoxic chemotherapy and could not be evaluated in culture due to high cytotoxicity even at the lowest concentration of 0.625 µM. The results of our screen are corroborated by previous reports of these drugs’ in vitro efficacy. Pibrentasvir has been demonstrated to inhibit the nsp14/nsp10 exonuclease of SARS-CoV-2 and to restrict the infection of Calu-3 cells with a reported EC50 of 0.7 µM [30]. Similarly, atovaquone has a reported EC50 of 1.5 µM in VeroE6 cells, 2.7 µM in VeroE6 cells expressing hTMPRSS2, and 6.8 µM in Huh7.5 cells but a markedly increased EC50 of 29.7 µM in Calu-3 cells [31,32].

Notably, atovaquone and pibrentasvir have differential effects on SARS-CoV-2 focus formation (Figure 3B). Compared to the untreated cells, both atovaquone and pibrentasvir limit the cell-to-cell spreading of SARS-CoV-2 as indicated by the reduced size and frequency of distinct foci. Foci in the atovaquone-treated cells stain for the spike protein with a similar intensity to the untreated cells; however, the foci in the pibrentasvir-treated cells stain more faintly. This may indicate that, in addition to preventing cell-to-cell spread, pibrentasvir disrupts earlier steps in viral replications leading to a reduced spike protein production. This observation agrees with the earlier described mechanism of nsp14/nsp10 exonuclease inhibition [30]. The proposed action of pibrentasvir and atovaquone on targets besides nsp15 is further supported by the high antiviral effect at concentrations where no nsp15 inhibition is observed in the fluorescence assay. These suggest that while the inhibition of nsp15 may augment the antiviral effect for each drug, it is unlikely to be the dominating mechanism for either.

### 3.3. Nsp15 Inhibitors Restrict HCoV-OC43 Infection In Vitro

As nsp15 is highly conserved among all mammalian coronaviruses, we sought to determine whether the inhibitory action of pibrentasvir and atovaquone would also be conserved against a related mild coronavirus, hCoV-OC43. As hCoV-OC43 does not depend on ACE2 for cellular entry, infections were performed in wild-type A549 cells and detected by staining with an anti-OC43 N protein monoclonal antibody. Both pibrentasvir and atovaquone inhibited hCoV-OC43 infection in vitro (Figure 4A). Furthermore, this inhibition could be detected by flow cytometry as early as 12 h post-infection (Appendix A). While pibrentasvir inhibited hCoV-OC43 to a similar extent as SARS-CoV-2, atovaquone had a reduced peak inhibition at higher concentrations but remained active at lower concentrations compared to SARS-CoV-2.

After confirming both drugs remained active against hCoV-OC43, we further examined whether they would lose inhibitory activity in a MAVS knockout cell line. Since MAVS acts as a downstream-signaling intermediary of the two cytosolic dsRNA sensors MDA5 and RIG-I, compounds which inhibit viral infections via nsp15 should be MAVS-dependent. Indeed, pibrentasvir and atovaquone experienced moderate and severe reductions in inhibitory activity, respectively (Figure 4A). The more modest reduction in inhibition for pibrentasvir suggests that this drug is only partially dependent on nsp15 and is governed by other mechanisms such as the previously discussed exonuclease inhibition [30]. As atovaquone lost inhibitory activity in the MAVS^−/−^ A549 cells, we hypothesized that atovaquone may alter the localization or accumulation of viral dsRNA. Previously, Deng et al. reported that an nsp15-deficient variant of MHV possessed normal amounts of dsRNA but altered localization, with an increase in cytosolic dsRNA not associated with replication transcription complexes [10]. We therefore investigated, by immunofluorescence, the localization of dsRNA in MAVS^−/−^ A549 cells with and without inhibitor treatment (Figure 4B,C). dsRNA staining was specific to the HCoV-OC43-infected cells with enrichment of the dsRNA foci in the perinuclear space typical of coronaviruses. Notably, atovaquone-treated cells displayed a greater number of distinct dsRNA foci per cell that spread throughout the cytoplasm, while pibrentasvir-treated cells were not significantly altered in dsRNA content.

Again, both pibrentasvir and atovaquone remain active against HCoV-OC43 at concentrations below their observed inhibition of nsp15 in the fluorescence assay. While it is possible these drugs may bind more tightly to the nsp15 of HCoV-OC43 than SARS-CoV-2, it is more likely that additional targets are present which together produce the antiviral effect. However, the effect of atovaquone does appear restricted to the sensing of dsRNA and further involves the localization of dsRNA. It is therefore plausible that, in addition to nsp15 inhibition, atovaquone is further augmenting the activation of cytosolic dsRNA sensors or potentially disrupting the formation of DMVs responsible for sequestering viral dsRNA.

### 3.4. Atovaquone and Pibrentasvir Are Not Protective in a Mouse Model of SARS-CoV-2 Infection

Following the in vitro characterization of atovaquone and pibrentasvir, we further examined whether either drug could limit SARS-CoV-2 infection in mice using a prophylaxis treatment regimen (Figure 5A). C57BL/6 mice were dosed with either 40 mg/kg of atovaquone or 24 mg/kg of pibrentasvir via daily intraperitoneal (i.p.) injections, beginning one day prior to infection with 1 × 10^6^ pfu of SARS-CoV-2 B.1.351 delivered intranasally. The dosing regimens were designed following the guidance of previous pharmacokinetic and antiviral studies of atovaquone and pibrentasvir in murine models [31,33,34,35]. Mice were monitored for weight loss daily beginning on day 0 until euthanasia at the peak of infection on day 3. RNA was extracted from the lung tissue and turbinates of each mouse to quantitate the extent of viral replication by RT-qPCR.

Despite the observed in vitro inhibition of the SARS-CoV-2 infection, neither atovaquone nor pibrentasvir was protective in the in vivo model. Although both RNA-dependent RNA polymerase (RdRp) and subgenomic (Sg) viral transcripts trended lower in the pibrentasvir- and atovaquone-treated groups, this effect was not statistically significant in either the lung or nasal turbinate tissues (Figure 5B–E). Moreover, there was no therapeutic effect, as the treated and control mice had nearly identical weight losses on day 3 post-infection (Figure 5F).

## 4. Discussion

Here, we report the discovery of nascent nsp15 inhibitory activity present in several drugs approved for human use by the USFDA for the treatment of conditions unrelated to SARS-CoV-2. These include idarubicin, an anthracycline inhibitor of DNA topoisomerase II used in the treatment of leukemia; pibrentasvir, an inhibitor of the hepatitis C virus nonstructural protein 5A; and atovaquone, an antimicrobial drug used in the treatment of several fungal and parasitic infections. While these drugs were not protective in our mouse model system, it remains possible that an alternative formulation, dosage, or delivery method, such as the direct application of the compound to airways via a nebulizer, could produce improved in vivo protection. The identification of these compounds and their in vitro characterization may serve as a basis for the future design of novel nsp15 inhibitors based on chemical modifications of the afore-mentioned drugs. Furthermore, the success of FRAGSITE2 in identifying pibrentasvir as a high-precision, highly ranked hit exemplifies the potential of this tool in future drug discoveries. One major advantage of FRAGSITE2 over the FINDSITE^comb2.0^ [36] and FRAGSITE [19] methods is that FRAGSITE2 does not use template ligands to derive the profile for the protein target; thus, it can make predictions for ligands which do not have close homologous template ligands in the PDB. Compared to the FINDSITE^comb2.0^ and FRAGSITE methods that depend on template ligands for deriving the target profile, FRAGSITE2 has a comparable performance while having the potential to discover ligands that are remote from the PDB, DrugBank, and ChEMBL libraries [37].

Until recently, the repertoire of known nsp15 inhibitors was essentially limited to the generic RNase A inhibitors benzopurpurin B and Congo Red whose activity was originally characterized by Ortiz-Alcantara et al. in 2010 [29]. Since then, five publications have expanded this list to include Tipiracil [24], NSC95397 [38], Exebryl-1 [39], epigallocatechin gallate [40], and betulonic acid derivatives [41]. Of these compounds, only the betulonic acid derivatives show substantial inhibition (<10 µM) of viral replication in cell cultures; however, this effect was restricted to HCoV-229E and was not reproducible for SARS-CoV-2, MHV-A59, or the feline infectious peritonitis virus (FIPV) [41]. Our findings complement these earlier studies and expand the repertoire of nsp15 inhibitors with in vitro efficacy. However, it should also be noted that both atovaquone and pibrentasvir possess additional mechanisms of SARS-CoV-2 inhibition, and it is unclear whether the interaction with nsp15 alone is sufficient to achieve their low micromolar effective range [30,32].

While pibrentasvir has not previously received a significant in vivo characterization for the treatment of COVID-19, two prior studies have examined atovaquone. In the first of these studies, Ahmed et al. performed a screen of predicted SARS-CoV-2 main protease inhibitors, which included atovaquone. Although atovaquone only modestly inhibited the main protease activity at 50 µM, it successfully inhibited viral replication with an IC50 of 1.5 µM in Vero E6 cells and 6.8 µM in Huh7.5 cells. However, atovaquone binds significantly to serum proteins in tissue culture medium and switching to a serum-free culture system further reduced the IC50 to just 20 nM in WHO Vero cells. They further determined the pharmacokinetics of atovaquone in a mouse model, showing an 18–38 h half-life and peak total plasma, total free plasma, and lung epithelial lining fluid concentrations of 273 µM, 44 nM, and 70 nM, respectively, after 7 days of receiving 20 mg/kg of atovaquone orally [31]. Although we did not quantitate the free plasma atovaquone in our dosing regimen, our regimen used double the daily dose and was not limited by the low bioavailability of atovaquone achieved by oral dosing [42]. The second study was a small randomized, double-blind, placebo-controlled trial to evaluate the efficacy of 1500 mg of oral atovaquone given twice daily to reduce the viral load over 10 days when first administered within 72 h of hospitalization [43]. This dosing regimen follows similar guidelines in the treatment and prevention of *Pneumocystis jiroveci* and *Pneumocystis carinii* pneumonia in adults, albeit the doses are doubled from 750 mg to 1500 mg twice per day. This provides an estimated daily dose in a similar range to our 40 mg/kg regimen, although the oral route likely reduces the bioavailability compared to i.p. injection. Despite a similar dose to both our study and the study of Ahmed et al., the total plasma concentration of atovaquone in the clinical trial peaked at 31.6 µM after 5 days of administration. Throughout the course of the clinical trial, atovaquone treatment was not found to significantly lower the viral load at any point during the 10 days of atovaquone treatment [43]. As a whole, these results reaffirm that even under the ideal conditions of prophylaxis and a high daily dose in our mouse model, atovaquone is ultimately ineffective at controlling SARS-CoV-2.

The experiments presented here provide some insight into the mechanism of action and in vivo potential of atovaquone and pibrentasvir but are ultimately limited in several regards. First, our in vivo study was based on dosing regimens previously used in pharmacokinetic studies or other disease models for which we have not determined the exact serum or lung epithelial fluid concentrations according to our dosing regimen and delivery method. It may be that a full dose escalation study in mice could identify a higher “safe” threshold for atovaquone or pibrentasvir treatment that would produce an in vivo protective effect. Second, our in vitro studies are limited by a lack of established assays for confirming the specific inhibition of nsp15 in the cellular context. To this end, we employed MAVS^−/−^ cells and the immunofluorescence of viral dsRNA puncta to make inferences regarding the activity of atovaquone and pibrentasvir in cell cultures, but we ultimately lack a robust and high-throughput assay to specifically evaluate nsp15 activity during cellular infection. The field of nsp15 research would greatly benefit from the identification of a compound that robustly inhibits nsp15 in cell cultures to act as a positive control and an assay which produces an easily quantifiable readout. This would allow researchers to skip the in vitro screens with recombinant nsp15 and simultaneously avoid pursuing inhibitors which do not function in the cellular context.

To conclude, the studies presented here show the first evidence of SARS-CoV-2 nsp15 inhibition for idarubicin, atovaquone, and pibrentasvir. Further characterization of the specific binding interactions between each compound and nsp15 will allow for the determination of the minimal structural elements required for nsp15 inhibition. These future studies will also provide the opportunity to enhance nsp15 binding specificity via the subtraction, substitution, or addition of chemical moieties to the base compound. Finally, our in vivo experiments answer long-standing questions concerning the clinical utility of atovaquone and pibrentasvir in COVID-19 treatments that were raised by in vitro experiments in prior publications.

## Figures and Tables

**Figure 1 viruses-15-01841-f001:**
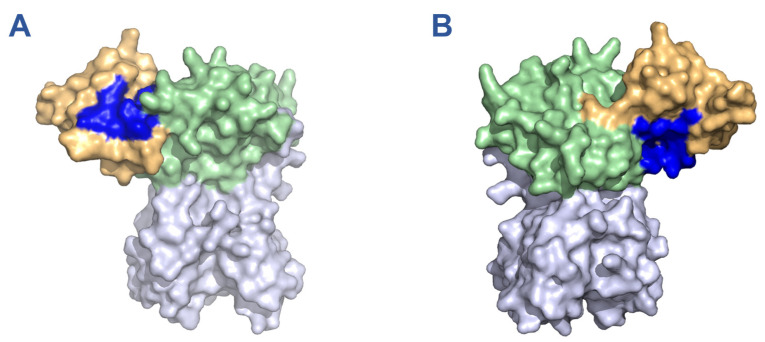
Binding pockets of 11 drugs on monomeric nsp15 predicted by in silico screening with FRAGSITE2. Binding pockets are highlighted in blue while the oligomerization, middle, and catalytic domains are depicted in orange, green, and purple, respectively. (**A**) Binding pocket 1, predicted for oritavancin, ledipasvir, posaconazole, micafungin, linaclotide, pibrentasvir, desmopressin, and cyanocobalamin. (**B**) Binding pocket 2, predicted for rifamixin, rifapentine, and everolimus. Depictions generated based on PDB structure 7N06.

**Figure 2 viruses-15-01841-f002:**
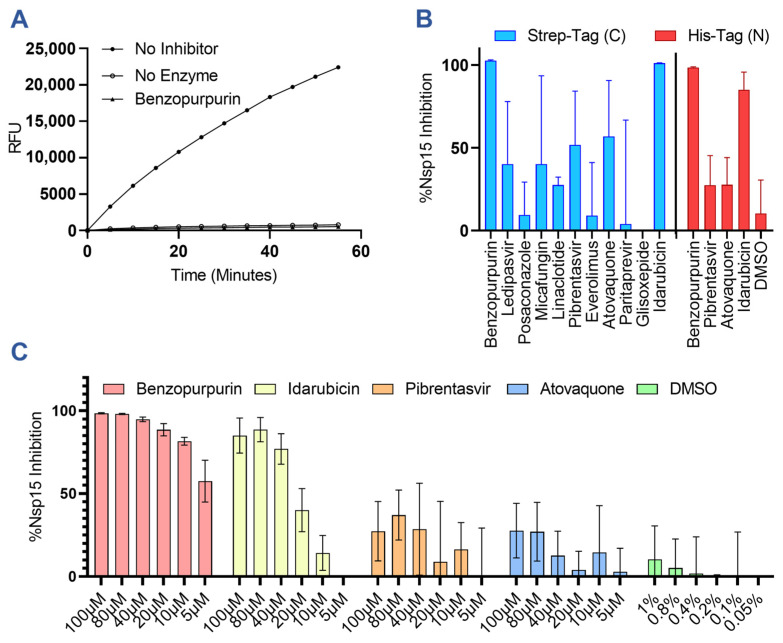
In vitro screening results of predicted inhibitors against nsp15 nuclease activity. (**A**) Representative results of a real-time nsp15 nuclease assay demonstrating cleavage of the FRET probe as measured by relative fluorescence units (RFUs) and quenching of the nsp15 nuclease activity by the addition of 100 µM of benzopurpurin. (**B**) Graph showing the percent inhibition of nsp15 nuclease activity by in silico predicted drugs at 100 µM for C-terminally strep-tagged nsp15 incubated at 30 °C or N-terminally his-tagged nsp15 incubated at 37 °C. (**C**) Dose–response curves for nsp15 inhibition at the indicated drug concentrations with N-terminally his-tagged nsp15. At 100 µM of the drug, the resulting solution contains 1% DMSO. Columns and error bars represent the average and standard deviation of 3 independent experiments.

**Figure 3 viruses-15-01841-f003:**
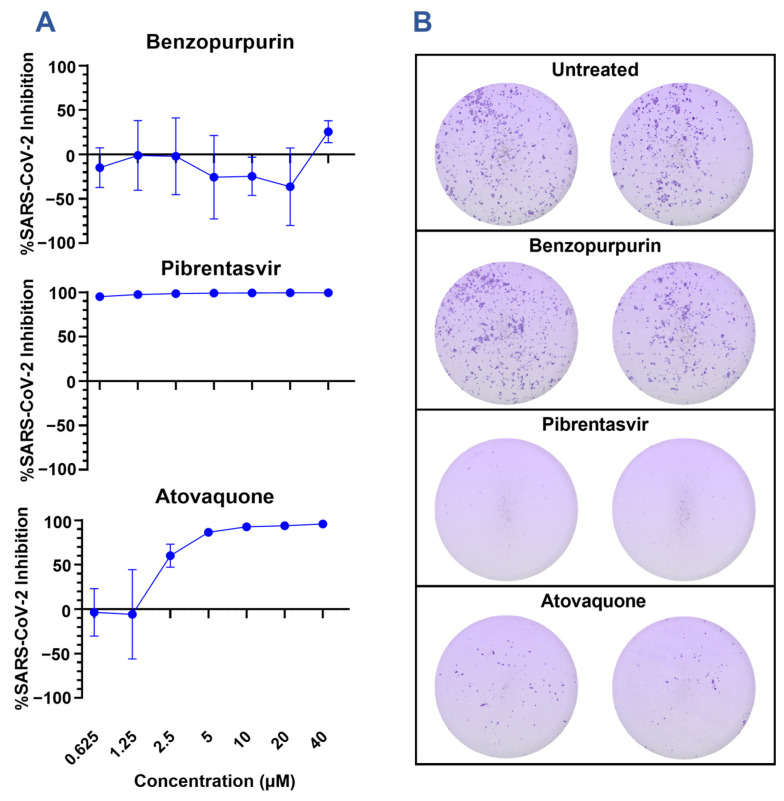
Inhibition of SARS-CoV-2 infection by in silico predicted inhibitors. (**A**) A549 hAT cells were infected with an MOI of 0.1 of SARS-CoV-2 A.1 (nCoV/USA_WA1/2020) for 2 h before the addition of select inhibitors at the indicated concentration. Foci were developed at 48 h using a cross-reactive anti-SARS-CoV-1 spike antibody, and the viral inhibition was calculated relative to the area of infection in the control untreated samples. Graphs are representative of 2 experiments performed in triplicate. (**B**) Representative images from focus forming assays are presented in 2A of samples treated with 10 µM of the indicated drug.

**Figure 4 viruses-15-01841-f004:**
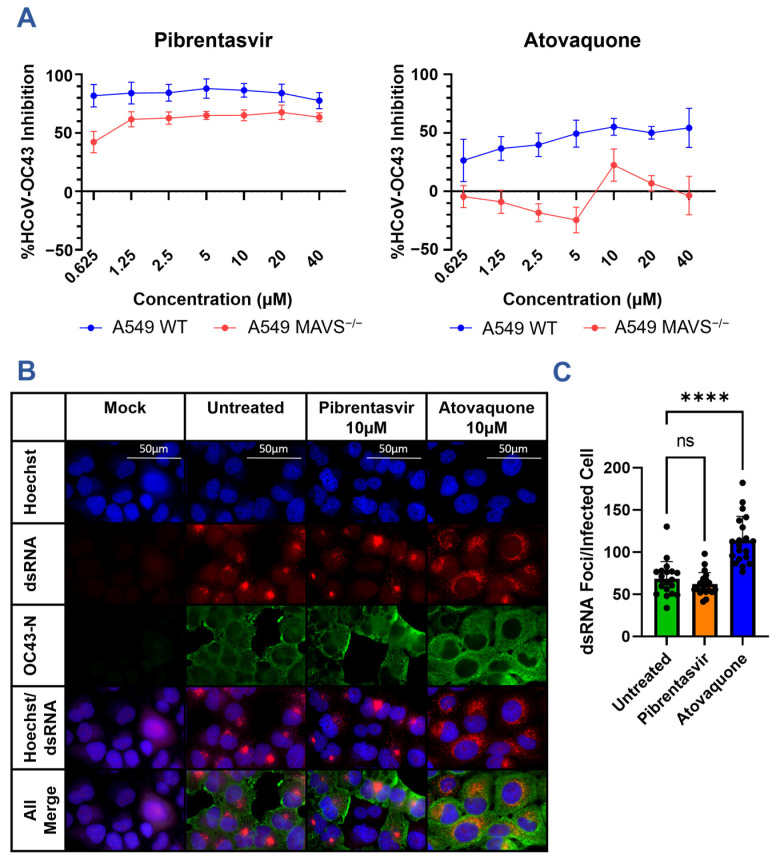
Inhibition of HCoV-OC43 infection by atovaquone and pibrentasvir. (**A**) A549 WT or A549 MAVS^−/−^ cells were infected at an MOI of 0.1 of HCoV-OC43 for 2 h before the addition of select inhibitors at the indicated concentrations. Foci were developed at 24 h using an anti-OC43 N protein antibody, and the viral inhibition was calculated relative to the area of infection in the control untreated samples. Graphs are representative of 2–3 experiments performed in triplicate. (**B**) Representative images of HCoV-OC43 infection following treatment with nsp15 inhibitors. A549 MAVS^−/−^ cells coated on glass slides were infected with HCoV-OC43 at an MOI of 0.1. Nsp15 inhibitors were added to the indicated concentrations 2 h post-infection. Slides were fixed 24 h post-infection, permeabilized, and stained for the presence of the HCoV-OC43 N protein and viral dsRNA with monoclonal antibodies. (**C**) Graphical representation of the number of distinct dsRNA foci per infected cell in 4B. dsRNA foci were automatically counted in ImageJ version 1.53f51. This plot is representative of 20 random fields of view taken from 5 experiments, where each field of view is represented by a black dot. Results analyzed by One-Way ANOVA with multiple comparisons. *p*-values represented as ns = not significant, **** < 0.0001.

**Figure 5 viruses-15-01841-f005:**
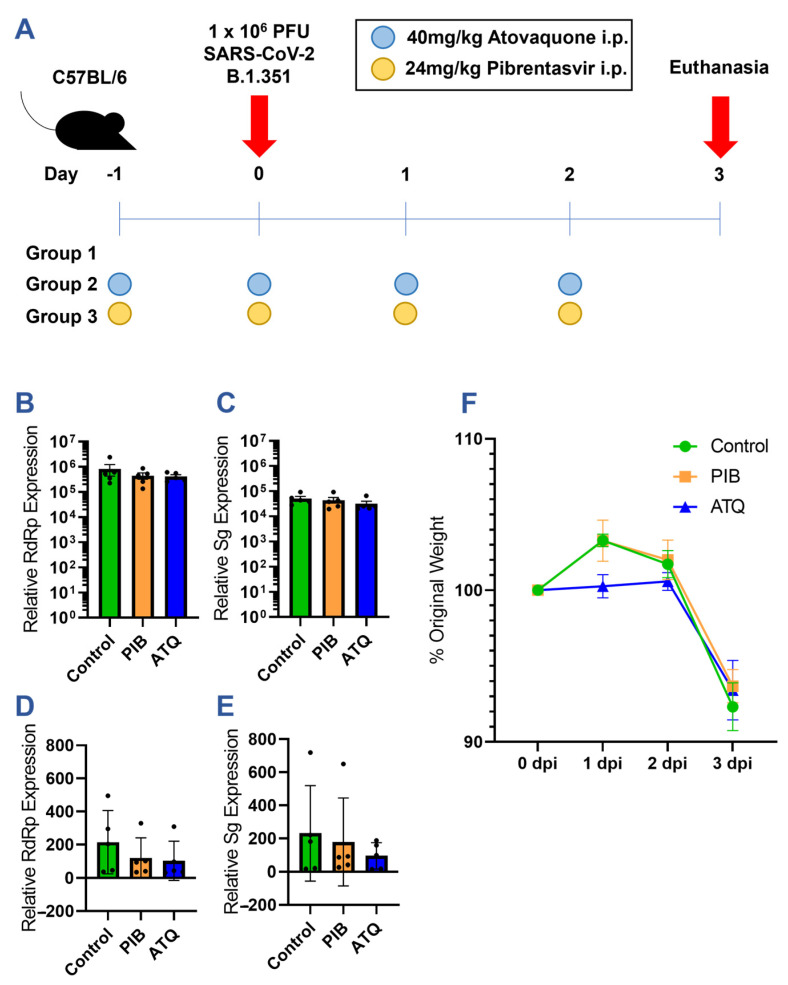
Efficacy of prophylactic atovaquone and pibrentasvir treatment in a mouse model of SARS-CoV-2 infection. (**A**) Diagram of mouse study. C57BL/6 mice were divided into 3 study groups with 5 mice per treatment regimen. (B–E) RT-qPCR quantitation of viral RNA copies in mouse lung and turbinate extracts: (**B**) Lung RdRp, (**C**) Lung Sg, (**D**) Turbinate RdRp, and (**E**) Turbinate Sg. Equal amounts of lung and turbinate cDNA from each mouse collected on day 3 was amplified with viral RdRp, viral subgenomic, and mouse GAPDH-specific primer/probe sets. The concentration of viral RNA is normalized across samples to GAPDH and expressed relative to the background amplification in uninfected lung tissue and turbinate samples. Black dots indicate values for individual mice. (**F**) Weights of mice in each treatment group normalized to the day 0 weight.

**Table 1 viruses-15-01841-t001:** Ranking of FDA-approved drugs predicted by FRAGSITE2 to bind nsp15 from SARS-CoV-2. Compounds are grouped by binding pocket 1 (orange) or binding pocket 2 (blue).

Drug Name	Score	Precision	Binding Site
Oritavancin	1.29	0.9	2LEU 26ILE 31VAL 50PRO 51VAL 52ASN 53VAL 55PHE 56GLU
Ledipasvir	1.10	0.9	
Posaconazole	1.06	0.9	
Micafungin	1.06	0.9	
Linaclotide	1.02	0.9	
Pibrentasvir	0.97	0.87	
Desmopressin	0.97	0.86	
Cyanocobalamin	0.93	0.83	
Rifamixin	0.99	0.88	31VAL 42LEU 43PHE 44GLU 46LYS 54ALA 55PHE 58TRP 59ALA 61ARG 86TRP 91ASP
Rifapentine	0.80	0.68	42LEU 43PHE 44GLU 46LYS 54ALA 55PHE 58TRP 59ALA 61ARG 86TRP 91ASP
Everolimus	0.77	0.65	

**Table 2 viruses-15-01841-t002:** Additional compounds predicted by other in silico studies to bind nsp15 from SARS-CoV2.

Drug Name	Binding Site	Reference
Atovaquone	Catalytic Pocket	[27]
Paritaprevir		[27]
Glisoxepide		[28]
Idarubicin		[28]

## Data Availability

The data presented in this study are contained within the article and Appendix A.

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
