# Peer review of "Atovaquone and Pibrentasvir Inhibit the SARS-CoV-2 Endoribonuclease and Restrict Infection In Vitro but Not In Vivo"

_viruses, 2023, doi:10.3390/v15091841_

Round 1
Reviewer 1 Report
I think the manuscript is interesting and helpful in the practice of new drug research and development for COVID-19 treatment. I have minor comments as follows:
1. Check the abbreviations throughout the manuscript and introduce the abbreviation when the full word appears the first time in the abstract and the remaining for the text and then use only the abbreviation. For example, “infectious bronchitis virus (IBV), murine hepatitis virus (MHV)”.
2. Please cite relevant literature. For example, “Severe acute respiratory syndrome coronavirus 2 (SARS-CoV-2), the causative agent of coronavirus disease of 2019 (COVID-19) continues to spread globally and cause significant morbidity and mortality among unvaccinated individuals and those with weakened immune systems.(Viruses 2023, 15(7), 1508)”; “This approach can accelerate the drug discovery pipeline by limiting the selection of potential therapeutics to those with known safety and tolerability profiles.( DOI: 10.1016/j.ejmech.2023.115503)”.
3. “Since then, 5 publications in 2021 expanded this list to include Tipiracil [22], NSC95397 [33], Exebryl-1 [34], epigallocatechin gallate [35], and betulonic acid derivatives [36].” There are several recent reports which are not included in this manuscript. Authors should update the manuscript by adding recent studies.
4. Nsp15 is a novel drug target to treat COVID-19. But the structural and functional information gained from SARS-CoV-2 nsp15 structure could be more concise.
5. In lines 360-362, In addition to lead optimization, please add “small molecule therapeutics have demonstrated potential value and with the assistance of nanotechnology, combination drug therapies, and the prodrug strategy, this remedy could serve as a starting point for further drug development in treating these lung diseases.(DOI: 10.3390/biomedicines9060689)”.
Minor editing
Reviewer 2 Report
The paper is well structured, discussing a very hot topic. I recommend this article to be published in the journal. Here are some suggestions:
1. Often pharmacological activities are reported very naively, and the authors need to provide a more critical assessment of what the data you report show and what not (i.e. limitations, problems in the design of the studies and future research needs. The same is in essence true for clinical studies.
2. There is a lack of recent literature citations.
3. The MS still needs very careful editing for scientific clarity. Importantly, this also includes the introduction and discussion.
4. The introduction begins by pointing out all of the failings of the effective of currently approved small molecule therapeutics. It is suggested to add some details. “remdesivir (DOI: 10.3389/fimmu.2022.1015355), molnupiravir (Nat. Struct. Mol. Biol., 28 (2021), 740-746), and ritonavir-boosted nirmatrelvir (Paxlovid) (DOI: 10.1002/jmv.27517)”
5. Significant progress has been made in treatment of COVID-19, but currently approved small molecule therapeutics have suffered from lack of global access, limited administration routes, and poor efficacy against SARS-CoV-2 variants (Eur. J. Med. Chem. 257 (2023), 115503). What about all the many failings of the MS? and issues regarding efficacy and safety?
Minor editing of English language required
Reviewer 3 Report
In this article, von Beck et al use in silico, in vitro, and in vivo methods to identify potential repurposed inhibitors of nsp15, a conserved nuclease found in coronaviruses. While the virus has mutated to escape and/or lessen the impact of the original vaccine and monoclonal antibodies, remdesivir and other drugs targeting the conserved replication transcription complex components remain effective. Thus, targeting other nsps could be a useful strategy. Multiple studies have attempted to identify effective inhibitors of nsp15 but have not yet been successful. Von Beck et al join these ranks. However, the inhibitors presented here could be backbones for more effective candidates. The combination of testing candidate repurposed inhibitors at multiple levels makes this worthwhile for publication, however there are issues with the in silico and in vitro work that need addressed prior to publication.
Major comments
-
1) The identification of target inhibitors section is lacking important details—for example, was the structural model of nsp15 used to identify inhibitors in the hexameric or monomeric form? Furthermore, it is not clear why a structure prediction generated by TASSER was used to identify inhibitors given there are numerous experimental structures of nsp15 across different coronaviruses in the PDB. The authors need to justify this decision.
-
2) The authors should clarify how many target pockets were considered in their search. Was the nsp15 active site used as the target pocket? Were other sites identified on the protein? If multiple pockets were screened, Tables 1 and 2 would be improved with a column that names the predicted pocket for each drug.
-
3) Following points 1 and 2, the paper needs figure panels detailing the predicted binding pose of the top candidate drugs, especially atovaquone and pibrentasvir. These figures will enable readers to understand the important chemical functional groups of the inhibitors and potential mechanisms of action. The authors state multiple times that although these drugs were not effective in vivo, the study is still useful because they might spawn new inhibitors based on these backbones. However, without details about where these bind that stated goal is undercut.
-
4) The protein purification method does not incorporate a step to selectively obtain hexameric nsp15, which has been shown to be the active form of the enzyme in vitro. The Western blot in Supp Fig 1 shows the purified protein is largely (perhaps >90%) monomeric. Further, the full blot shows there appears to be variability in the amount of hexamer obtained across different purifications. Were there any steps taken to normalize the amount of hexamer used in the FRET nuclease assay? Otherwise the % inhibition could be biased by different amounts of monomer in the enzyme sample—perhaps this explains in part the very large error bars seen in Figure 1B. Additionally, the tag is not cleaved off, and as a C-terminal tag, it may interfere with the EndoU domain.
-
5) The nuclease assay used 0.5 µM RNA, 30 nM enzyme, and an incredibly large excess of inhibitor (100 µM)—were lower concentrations of inhibitor tested? Why was 100 µM chosen when 40 µM is this highest concentration used in subsequent experiments? How are the inhibitors solubilized—do they require DMSO? If DMSO is present, was the assay run with a DMSO control?
-
6) The authors should include controls for aggregation. Benzopurpurin will induce nsp15 aggregates and is not a true inhibitor. Check that the inhibitors tested are not simply reducing fluorescence in the activity assay by causing protein aggregation.
Minor comments
-
1) The use of italics for in silico, in vitro, and in vivo is inconsistent throughout the paper. Check that all instances are italicized.
-
2) The methods section is missing details on the composition of buffers (ex: elution buffer) as well as the pH of buffers (ex: NendoU buffer).
-
3) The methodology describes very different temperatures and incubation times for experiments with SARS-2 vs OC43. It is not clear why these parameters were chosen.
-
4) While Figure 4 describes how the mice were treated with inhibitors, this should also be verbally documented in the methods section of the paper. Additionally, how were the concentrations of inhibitors determined? This is alluded to but should be explicit.
-
5) Figure 3B mentions nsp15 inhibitors were added to the indicated concentration, but it is unclear where that is indicated. It would be best to add to the column labels.
-
6) If possible, given the multiple RTC targets of atovaquone and pibrentasvir, Fig 3B would benefit from a set of panels demonstrating the change in dsRNA foci in nsp15-deficient OC43.
-
7) The authors state that no new data were created or analyzed in this study, so data sharing is not applicable. Is table 1 not new data? Again, if the stated aim to share this study so people can potentially iterate new nsp15 inhibitors is made in good faith, the screen should be available.
Round 2
